# Genetic Algorithm for Solving the No-Wait Three-Stage Surgery Scheduling Problem

**DOI:** 10.3390/healthcare11050739

**Published:** 2023-03-02

**Authors:** Yang-Kuei Lin, Chen-Hao Yen

**Affiliations:** Department of Industrial Engineering and Systems Management, Feng Chia University, Taichung 407102, Taiwan

**Keywords:** scheduling, surgery, operating rooms, genetic algorithm, makespan

## Abstract

In this research, we consider a deterministic three-stage operating room surgery scheduling problem. The three successive stages are pre-surgery, surgery, and post-surgery. The no-wait constraint is considered among the three stages. Surgeries are known in advance (elective). Multiple resources are considered throughout the surgical process: PHU (preoperative holding unit) beds in the first stage, ORs (operating rooms) in the second stage, and PACU (post-anesthesia care unit) beds in the third stage. The objective is to minimize the makespan. The makespan is defined as the maximum end time of the last activity in stage 3. Minimizing the makespan not only maximizes the utilization of ORs but also improves patient satisfaction by allowing treatments to be delivered to patients in a timely manner. We proposed a genetic algorithm (GA) for solving the operating room scheduling problem. Randomly generated problem instances were tested to evaluate the performance of the proposed GA. The computational results show that overall, the GA deviated from the lower bound (LB) by 3.25% on average, and the average computation time of the GA was 10.71 s. We conclude that the GA can efficiently find near-optimal solutions to the daily three-stage operating room surgery scheduling problem.

## 1. Introduction

As medical technology improves day by day, more and more patients can be treated and saved through surgeries. Hence, operating rooms are in high demand in most hospitals. Operating rooms usually consume more than 9% of a hospital’s budget [1]. However, it is also estimated that operating rooms account for more than 40% of a hospital’s revenue [2]. Operating rooms are high-cost, high-revenue, and high-demand units. Finding an effective method for allocating surgeries to operating rooms that maximizes operating room utilization while ensuring good quality of care is a very important problem for hospital managers. In this research, we consider a daily surgery scheduling problem with three successive stages: pre-surgery, surgery, and post-surgery. The pre-surgery stage usually takes place in the PHU, where nurses help patients prepare for surgery (signing paperwork, putting on a hospital gown, performing necessary tests, measuring vital signs, setting up intravenous access, etc.). The surgery stage is the main stage of the entire operation, which takes place in the operating room. The post-surgery stage begins after the end of surgery and takes place in the PACU bed, where the patient is continuously monitored and cared for until recovery. The no-wait constraint is considered between the three stages. If no PACU bed is available after surgery, the patient remains in the operating room for recovery until a bed becomes available. Surgeries are known in advance (elective). Multiple resources are considered throughout the surgical process: PHU beds in the first stage; operating rooms in the second stage; and PACU beds in the third stage. The objective is to minimize the makespan. Figure 1 shows a typical patient flow in surgery and also shows the scope of this research.

Our main contribution is the development of an easy-to-implement GA, which efficiently solves the no-wait three-stage surgery scheduling problem. We also propose a heuristic and use an existing lower bound to validate the effectiveness of the proposed GA. The computational results show that the proposed GA can significantly reduce the makespan. Minimizing the makespan maximizes the utilization of ORs, reduces idle times between two consecutive surgeries, and also reduces material (PHU, operating rooms, PACU, equipment operating costs) and human (surgeons, nurses, anesthetists) costs. From the patient’s perspective, a reduction in time improves patient satisfaction. It can reduce waiting time and also increases the safety of the procedure since surgeons, nurses, and anesthetists working long overtime hours under high pressure are dangerous to patients during the procedure.

The remainder of this paper is organized as follows. Section 2 reviews the literature. Section 3 describes the problem. Section 4 presents the heuristic and the proposed GA. Section 5 presents the computational experiments and analyzes the results. Finally, conclusions and suggestions for future research are formulated. 

## 2. Literature Review

In the past 50 years, a large amount of literature on operating room scheduling has evolved. Refs. [3,4,5] provide a comprehensive survey of research on operating room planning and scheduling problems. Here, we review the literature pertaining to this study. Many researchers have studied operating room scheduling problems in a deterministic environment. Some researchers focused on studying the allocation of surgeries to operating rooms (stage 2) without considering the availability of human resources [6,7]. The objective is to minimize total operating costs, which include unused operating room costs and overtime costs. Some researchers considered both the availability of operating rooms and surgeons [8,9,10,11]. Some researchers studied two-stage (stages 2 and 3) operating room scheduling problems [12,13,14,15,16,17]. For example, references [12,13,14] considered scheduling surgeries in operating rooms and PACU beds (stages 2 and 3) with consideration of the availability of surgeons. The two-stage scheduling problem was further examined by references [15,16,17] by considering both human resources and equipment availability.

Only a few papers studied operating room scheduling problems in three stages [18,19,20,21]. In [18], the authors applied an ant colony (ACO) algorithm to solve a three-stage operating room scheduling problem integrating multiple nurses’ roster constraints. Similarly, in [19], the authors used an ACO to solve a three-stage operating room scheduling problem by considering available material resources (PHU beds, ORs, PACU beds) and human resources (surgeons, nurses, and anesthetists). Moreover, in [20], the authors proposed an iterative local search (ILS) and a hybrid GA (HGA) approach to solve the three-stage no-wait operating room surgery scheduling problem under various resource constraints. The availability of material resources, specialties, and qualifications of human resources was integrated, and the objective was to minimize the makespan and total idle time in the operating rooms. They compared the proposed ILS and HGA with real data and the ACO algorithm [19]. The ILS and HGA outperformed the real data scheduling and the ACO scheduling [19] with close results and a relatively short computation time. They allow fewer operating rooms and balance the working time between them. They concluded that the ILS and HGA algorithms are efficient and effective. In [21], the authors investigated operating room scheduling, taking into account all resources, post-anesthesia beds, and emergency procedures. To solve the problem, they formulated a mixed linear integer programming (MILP) and transformed it into CP models. They also proposed a metaheuristic based on a GA and a constructive heuristic (CH). They compared the heuristic (GA combined with CH) with the MILP and CP models. They concluded that the CP model discovered high-quality solutions faster than the MILP. However, the improvement was not substantial. On the other hand, the heuristic found solutions of very good quality in a short run time, with an average of 7 s, on instances ranging from 15 to 40 surgeries requiring scheduling. The heuristic outperformed the MILP and CP models. Table 1 presents comparisons between key related work and this study.

Some researchers tried to use mathematical model-based algorithms to solve operating room scheduling problems [6,8,15,16,17,21]. In [6], the authors developed a branch-and-price approach to solving a surgery assignment problem. In [8], the authors applied a column-generation-based heuristic to solve a tactical operating room planning problem. In [15], the authors proposed a primal–dual heuristic for solving an operation theatre planning problem. In [16], the authors presented a two-step approach to address an operating room scheduling problem with various resource-related constraints and the specifications of the operations processes. The two-step approach is based on solving mathematical models. In [17], the authors developed a constraint programming approach for solving multi-objective operating room scheduling. They considered multiple real-life constraints in the mathematical model, such as availability, staff preferences, and affinities among staff members. Mathematical model-based algorithms usually need advanced knowledge in math and optimization. It might not be easy to learn and implement for hospital managers. On the other hand, some researchers have recently addressed operating room scheduling problems using machine learning-based algorithms. Machine learning-based techniques are mainly employed to handle stochastic problems or problems with uncertain data. For example, [22] developed several machine learning techniques for the predictive energy consumption data of smart residential homes. For operating room scheduling problems, machine learning-based algorithms are applied to predict surgery durations, estimate required PACU time for each type of surgical procedure, predict emergency patient arrival times, or detect surgeries with significant cancellation risks [23,24,25,26,27]. Since we are dealing with a deterministic three-stage operating room surgery scheduling problem, machine learning-based algorithms might not be suitable. Additionally, several researchers have applied metaheuristics to handle operating room scheduling issues, including GA [7,12,20,21,28,29], ILS [20], ACO [18,19], and artificial bee colony (ABC) [10] algorithms. The advantage of metaheuristics is that they are easy to implement and can find good quality solutions within a reasonable time. This might better suit the needs of the hospital manager. In contrast to previous mathematical model-based approaches [6,8,15,16,17,21], we aim to develop an algorithm that is easy to implement and flexible for the majority of hospitals. GA is one of the oldest and most well-known algorithms that are easy to learn and implement. Hence, we have chosen GA to address the research problem. 

From the literature review, there is research studying the three-stage operating room surgery scheduling problem. All have considered both human resources (surgeons, anesthetists, nurses) and material resources (PHU, ORs, PACU). To create a feasible schedule, the availability of human resources and material resources must be integrated. However, human resources are generally very complicated and must take into account a variety of constraints, including resource availability (role, shifts, roster), specialties, qualifications, etc. This usually makes the developed algorithms and results only suitable for the specified problems. Hence, in this study, we only consider the three-stage operating room surgery scheduling problem under material resource constraints. We do not take into account human resources to maintain the developed algorithm’s simplicity and flexibility for the majority of hospitals.

In this research, we consider the PHU and PACU beds while constructing the operating room schedule. Similar to a three-stage no-wait (nwt) flexible flow-shop problem, the operating room scheduling problem under consideration aims to reduce the makespan. Following the three-field notation of Graham et al. [30], we refer to this problem as ${FF}_{3}|nwt|C_{max}$. The studied problem is NP-hard in the strong sense since the problem of minimizing makespan on a three-stage no-wait flow-shop problem ($F_{3}|nwt|C_{max}$) is already NP-hard in the strong sense [31]. Given that the studied problem is NP-hard, we have proposed a genetic algorithm (GA) to handle it.

## 3. Problem Description

This study deals with a daily surgery scheduling problem that is equivalent to a flexible flow shop problem with three sequential stages: preoperative (pre-surgery), intraoperative (surgery), and postoperative (recovery). For each surgery, different resources at each stage are required: a PHU bed at the pre-surgery stage, an operating room at the surgery stage, and a PACU bed at the recovery stage. The no-wait constraint is considered between the three stages. Surgeries are known in advance (elective). The performance measure is to minimize the makespan. The makespan is defined as the maximum end time of the last activity in stage 3: $\min C_{max}=max(C_{31},C_{32},C_{33}\ldots C_{3n})$, where $C_{3j}$ is the completion time of surgery *j* on stage 3. We provide the following illustration of the examined problem using an example.

### Example A

We use a 10-surgery example to illustrate a three-stage operating room surgery scheduling problem. Material resources are 2 PHU beds, 3 ORs, and 2 PACU beds. The planning horizon is one day. Table 2 shows the data on surgeries.

Figure 2 depicts a feasible solution 1 for example A. The first surgery starts at 8:00, and the final surgery ends at 16:30. No-wait constraints among the three stages are met. The makespan is 16:30. As we can see, the operating rooms are idle for 5 h and 45 min. This schedule is not ideal. The expense of operating rooms is expensive. Any idle time in the operating room between two surgeries is a waste. Moreover, in order to avoid any delays, patients typically arrive at the hospital much early than scheduled. From the perspective of the patients, if procedures 9 and 10 can start earlier and finish sooner, it can cut down on waiting time and allow patients to go back to the ward (or home). Figure 3 shows a feasible solution 2 for example A. The first surgery starts at 8:00, and the final surgery ends at 15:30. No-wait constraints among the three stages are satisfied. The makespan is 15:30. Feasible solution 2 provides a better solution that has a smaller makespan and no idle times between two consecutive surgeries in an operating room. All other surgeries end before 13:30 except for surgery 8, which ends at 15:30. Minimizing makespan not only maximizes the utilization of ORs, reduces idle times and costs but also improves patient satisfaction. It can reduce waiting time so patients can return to the ward (or home) earlier and recuperate. This is a small example, with 10 surgeries to be scheduled. The manager might be able to manually optimize it. However, as the problem size increases, the complexity of the problem increases. Manual schedule optimization becomes virtually impossible. In that case, an easy-to-implement algorithm that can provide a good quality solution quickly would be highly beneficial. 

## 4. GA for a Three-Stage Operating Room Scheduling Problem

The GA was created by John Holland in 1960 [32]. GA is an adaptive technique that imitates the behavior of natural selection, evolution, and heredity. Although GA was created 60 years ago, it remains one of the most well-known and effective algorithms for solving various kinds of optimization problems. Readers interested in learning more about GA and its applications may refer to [33,34]. Many studies have successfully employed GAs to solve scheduling problems in operating rooms [7,12,20,21,28,29]. The literature review demonstrates that GA can effectively address scheduling problems for operating room surgeries and deliver positive outcomes. GA is a good fit because we are dealing with a three-stage deterministic operating room surgery scheduling problem [20,21]. As a result, we created a GA to address the research challenge. 

We consider giving the GA a good initial solution to guide the GA’s search in a good direction. Hence, a solution based on the longest processing time first (LPT) heuristic is developed. The LPT heuristic is described below.

### LPT Heuristic

Step 1. Let *U* be the set of unscheduled surgeries.

Step 2. Set $U=\left\{null\right\}$. Sum the durations of three stages for each surgery *j* ($\sum_{i}^{3}p_{ij}\forall j$). Add all unscheduled surgeries into *U* first and sort the surgeries in non-increasing order of $\sum_{i}^{3}p_{ij}$ values.

Step 3. If *U* = {null}, terminate the procedure and report the final makespan value. Otherwise, select the first surgery $j^{*}$ in *U*. Assigns job $j^{*}$ to each stage. In each stage, surgery $j^{*}$ is assigned to the first available resource (a PHU bed in the first stage, an operating room in the second stage, and a PACU bed in the third stage). The no-wait constraint has to be satisfied among the three stages.

Step 4. Remove $j^{*}$ from *U* by setting *U = U\*$j^{*}$ and go back to Step 3.

We describe the main steps of the proposed GA as follows.
1Coding and initial population

A solution is encoded in the proposed GA as a permutation of the *n* surgeries. The initial population contains an initial solution that was created using the LPT heuristic. The remaining population is created by randomly generating permutations of surgeries.
2Evaluation and selection

The GA uses a roulette wheel as the primary selection mechanism. The selection operates on an enlarged sampling space. The roulette wheel [33] gives a higher probability to the best individuals. A chromosome’s probability for each performance measure can be calculated from Equation (1), where $f\left(w\right)$ is the worst fitness value in the population, and $f\left(x\right)=C_{max}$ is the fitness value of a given solution x. Moreover, to preserve the best chromosome for the next generation, we always sort and keep the top 5 elite solutions and carry them into the next generation.
(1)$$P\left(x\right)=(f\left(w\right)-f(x))/\sum_{k=1}^{Pop}\left(f\left(w\right)-f(x_{k})\right)$$
3Crossover and mutation schemes

Two parent solutions are selected from the population based on a roulette wheel mentioned above for crossover. The crossover scheme is a two-point crossover operator [35], as depicted in Figure 4. The surgeries between two randomly selected points are always inherited from Parent-1 to the child. The rest of the chromosome is filled by reading the information of Parent-2 from left to right. This crossover is executed under a probability $P_{c}$. After crossover, there are $\left\lceil P_{c}\times Pop\right\rceil$ offspring generated.

The mutation happens in offspring under a probability $P_{m}$. The mutation scheme merely employs a general pair-wise exchange. Two surgeries are chosen at random, and then their positions are switched. The mutation scheme is illustrated in Figure 5.
4Termination criteria

There are two stopping criteria. One is based on the maximum number of iterations (MaxIteration), and the other one is based on the maximum number of iterations without improvement (MIWOI), whichever is earlier. If the GA runs MaxIteration or if all of the elite solutions do not improve for a pre-defined number (MIWOI), then we stop the GA and report the best solution.
5Parameterization

The performance of the GA is influenced by parameter settings. Here, we tune several important parameters of GA through extensive experiments. The final selected GA parameters are MaxIteration = 5000, Pop = 200, crossover rate $P_{c}$ = 0.75, mutation rate $P_{m}$ = 0.05, and MIWOI = 200. Figure 6 shows the search procedure for the proposed GA.

## 5. Computational Results

In this section, random problem instances are used to evaluate the GA’s performance. The GA and LPT heuristics were written in the Python language and run on a computer with an Intel core i7 2.3 GHz CPU and 16 GB RAM. Randomly generated problem instances were generated based on [19]. According to the surgery duration, the surgeries are classified into five types: small, medium, large, extra-large, and special. Table 3 shows the durations of pre-surgery, surgery, and post-surgery. Table 4 shows detailed information on four test cases, including the number of surgeries to be scheduled, the number of PHU and PACU beds, the number of operating rooms, and the number of surgeries in each surgery type. For each case, 10 randomly generated problem instances are generated and tested.

We calculate the lower limit (LB) of the tested problem instances in order to demonstrate the GA’s effectiveness. The lower bound is developed by [36] for solving a flexible flow shop to minimize the makespan. The studied problem has a no-wait constraint, so the lower bound developed for solving the flexible flow shop to minimize the makespan will also be the lower bound of the studied problem. Table 5 shows the results of the 10 surgeries under the 2 PHU beds, 3 ORs, and 2 PACU beds available. The average $C_{max}$ value of GA is 366.1 (minutes), which is about 6 h. If the first surgery starts at 8:00, then the last surgery would end at around 14:00, excluding lunch hours. Both the GA and the LPT can find solutions that finish 10 surgeries within 8 working hours.

Table 6 shows the results of 15 surgeries with 3 PHU beds, 4 ORs, and 3 PACU beds available. The average $C_{max}$ value of GA is 395.4 (minutes), which is about 6.5 h. If the first surgery starts at 8:00, then the last surgery can be finished before 15:00, excluding lunch hours. Both the GA and the LPT can find solutions that finish 10 surgeries within 8 working hours.

When the number of surgeries increases while the number of material resources is almost the same as in Case 2, overtime is needed slightly. Table 7 shows the results of 20 surgeries with 3 PHU beds, 4 ORs, and 4 PACU beds available. The average $C_{max}$ value of GA is 504.2 (minutes), and the largest $C_{max}$ value of GA is 532 (minutes), which are both more than 8 h. If the first surgery starts at 8:00, then the last surgery can be finished before 17:00, excluding lunch hours. However, 3 out of 10 surgeries cannot be finished within 8.5 working hours. Slight overtime is needed, and that causes extra costs.

On the other hand, the average $C_{max}$ value of LPT is 522.4 (minutes). The largest $C_{max}$ value of LPT is 543 (minutes), which is more than 9 h. A total of 8 out of 10 surgeries cannot be finished within 8.5 working hours. Overtime is more severe than the solutions found by the GA. Allowing more overtime can build up pressure and stress on the surgery team and increases the risk of surgery.

When the number of surgeries that needs to be scheduled is more than the capacity and overtime is needed, every minute counts. Table 8 shows the results of 30 surgeries with 4 PHU beds, 5 ORs, and 5 PACU beds available. The average $C_{max}$ value of GA is 604 (minutes), and the largest $C_{max}$ value from GA is 629 (minutes), which are both more than 10 h. If the first surgery starts at 8:00 am, then the last surgery would be finished around 19:00, excluding lunch hours. The overtime intensity increases. Patients are at risk during surgery when surgeons, nurses, and anesthetists work long hours under stress. Any potential mistakes might occur during the surgery. That increases the risk of surgery.

On the other hand, the $C_{max}$ found by the LPT is even worse than $C_{max}$ found by the GA. The average $C_{max}$ value from LPT is 623.6 (minutes). The largest $C_{max}$ value from LPT is 652 (minutes), which is almost 11 h. A shorter makespan improves patient satisfaction because it can arrange treatments for patients timely. A shorter makespan also improves the surgery team’s satisfaction because allowing more overtime can build up pressure and stress on the surgery team. The GA can save about 20 min on average when compared to the $C_{max}$ value found by the LPT. This will save costs, reduce the risk of surgery, and improve the satisfaction of patients and the surgery team.

Table 9 shows the overall results of the GA and the LPT, which is a summary of Table 5, Table 6, Table 7 and Table 8. The results indicate that the solutions found by the GA deviate by 3.27%, 4.53%, 2.54%, and 2.44% from the lower bound for 10, 15, 20, and 30 surgeries, respectively. On the other hand, the solutions found by the LPT deviate by 9.71%, 8.95%, 6.25%, and 5.74% from the lower bound for 10, 15, 20, and 30 surgeries, respectively. The GA outperforms the LPT in terms of $C_{max}$. However, the LPT outperforms the GA in terms of computation time. The overall average computation time of LPT is 0.05 s. The average computation time of the GA is 5.75, 8.88, 10.92, and 17.30 s for 10, 15, 20, and 30 surgeries, respectively. As we know, the optimal solutions would be values between the lower bound and upper bound (found by the GA). Hence, we conclude that the GA can find near-optimal solutions (within 3.2% deviation on average) efficiently.

## 6. Conclusions

In order to solve a daily three-stage operating room operation scheduling problem, we propose and develop an LPT heuristic and a GA. Four cases that consider the various surgical sizes under several material resource (PHU beds, ORs, and PACU beds) constraints are tested. A lower bound is calculated to evaluate the performance of the proposed LPT heuristic and the GA. The computational results show that the proposed GA outperforms the LPT heuristic in terms of makespan. Overall, the LPT is 7.66% deviated from the lower bound on average, while the GA is 3.20% deviated from the lower bound on average. The overall average computation times of the LPT and the GA are 0.05 and 10.71 s. We conclude that the daily three-stage operating room surgery scheduling problem can be efficiently solved by the GA. In many real-world problems, a surgery scheduling problem’s precise data might not be known in advance. The scheduling problem is stochastic because of unknowns regarding patient arrival and procedure length. Future work can focus on expanding the GA to address scheduling issues in operating rooms under uncertain conditions, such as unpredictable operation lengths, unpredictable cancellations, or unpredictable emergency patient arrivals.

## Figures and Tables

**Figure 1 healthcare-11-00739-f001:**
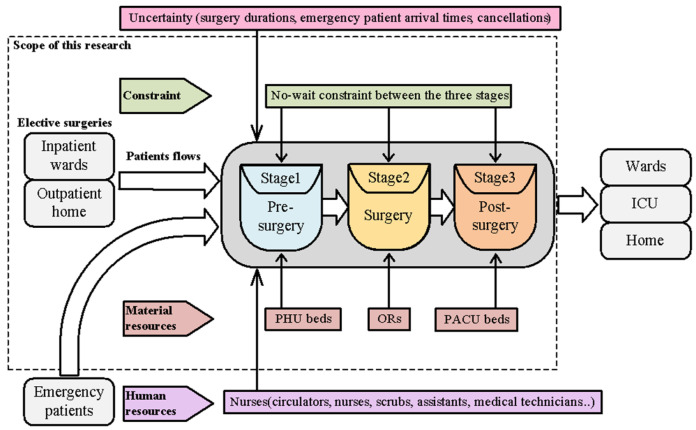
The scope of this research.

**Figure 2 healthcare-11-00739-f002:**
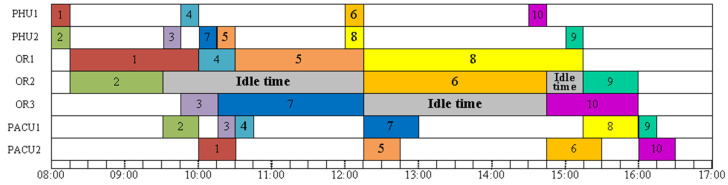
A feasible solution 1 for Example A.

**Figure 3 healthcare-11-00739-f003:**
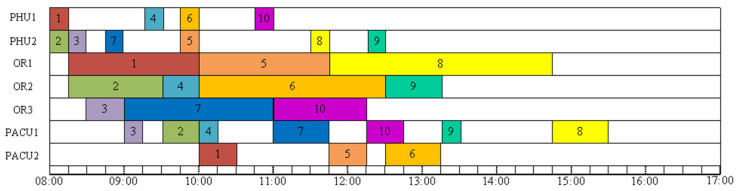
A feasible solution 2 of Example A.

**Figure 4 healthcare-11-00739-f004:**
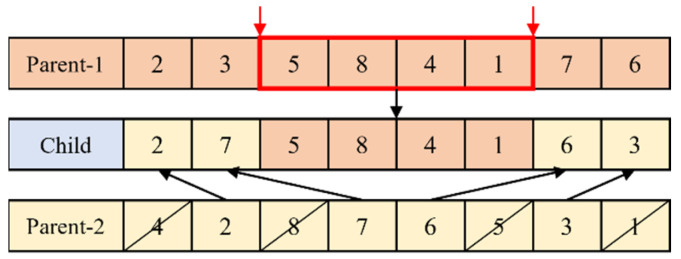
Crossover scheme.

**Figure 5 healthcare-11-00739-f005:**
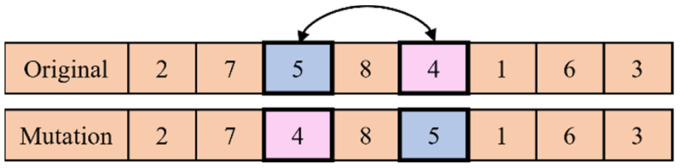
Mutation scheme.

**Figure 6 healthcare-11-00739-f006:**
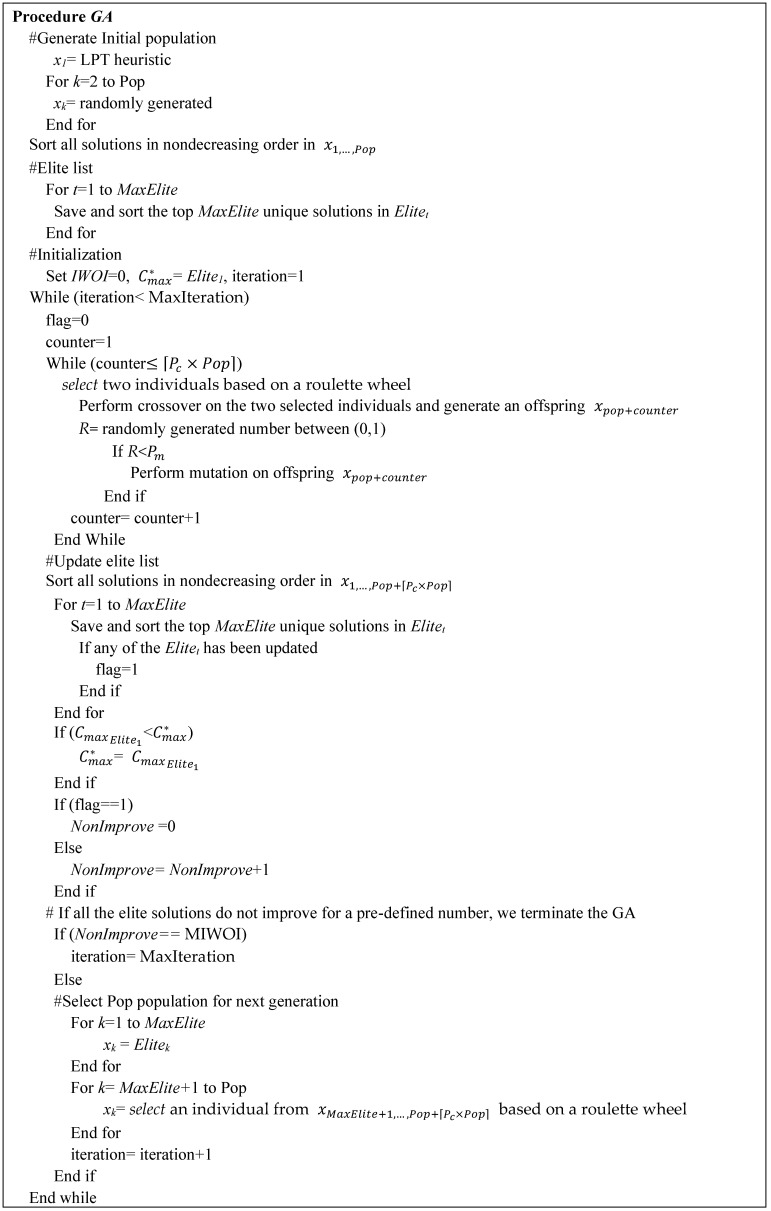
A search procedure for the proposed GA.

**Table 1 healthcare-11-00739-t001:** Comparisons between key related work and this study.

Paper	Stage	Objective	Algorithm	Analysis	Resource	Complexity
Fei et al. [6]	2	Minimize a cost function	Heuristic; mathematical model	A; E	ORs	NP-hard
Lin and Chou [7]	2	Minimize a cost function	Heuristic; mathematical model	A; E	ORs	NP-hard
Fei et al. [8]	2	Minimize a cost function	Heuristic	A	Surgeons, ORs	NP-hard
Zhu et al. [9]	2	Minimize a cost function	Heuristics	A	Surgeons, ORs	NP-hard
Lin and Li [10]		Minimize a cost function	Heuristic; mathematical model	A; E	Surgeons, ORs	NP-hard
Bargetto et al. [11]	2	Maximize the total surgery revenue	Heuristic; mathematical model	A; E	Surgeons, nurses, ORs	NP-hard
Fei et al. [12]	2, 3	Minimize a cost function	Heuristic	A	Surgeons, ORs, PACU	NP-hard
Liu et al. [13]	2, 3	Minimize a cost function	Heuristic	A	Surgeons, ORs, PACU	NP-hard
Riise et al. [14]	2, 3	Minimize makespan	Heuristic	A	Surgeons, ORs, PACU	NP-hard
Guinet and Chaabane [15]	2, 3	Minimize a cost function	Heuristic	A	Surgeons, ORs, equipment, PACU	NP-hard
Jebali et al. [16]	2, 3	Minimize a cost function	Heuristic; mathematical model	A; E	Surgeons, ORs, equipment, PACU	NP-hard
Meskens et al. [17]	2, 3	Minimize muti-objective	Constraint programming	A	Surgeons, nurses, anesthetists, ORs, equipment, PACU	NP-hard
Xiang et al. [18]	1, 2, 3	Minimize makespan	Heuristic; mathematical model	A	Surgeons, nurses, anesthetists, PHU, ORs, PACU	NP-hard
Xiang et al. [19]	1, 2, 3	Minimize makespan	Heuristic	A	Surgeons, nurses, anesthetists, PHU, ORs, PACU	NP-hard
Belkhamsa et al. [20]	1, 2, 3	Minimize makespan	Heuristic	A	Surgeons, nurses, anesthetists, PHU, ORs, PACU	NP-hard
Latorre-Núñez et al. [21]	1, 2, 3	Minimize makespan	Heuristic; mathematical model	A; E	Surgeons, nurses, anesthetists, ORs, equipment, PACU	NP-hard
This study	1, 2, 3	Minimize makespan	Heuristics	A	PHU, ORs, PACU	NP-hard

Stage: 1—considered pre-surgery, 2—considered surgery, 3—considered post-surgery. Analysis: A—approximate, E—exact.

**Table 2 healthcare-11-00739-t002:** Data of surgeries.

Surgery	Pre-Surgery Duration	Surgery Duration	Post-Surgery Duration
1	15	105	30
2	15	75	30
3	15	30	15
4	15	30	15
5	15	105	30
6	15	150	45
7	15	120	45
8	15	180	45
9	15	45	15
10	15	75	30
unit minutes			

**Table 3 healthcare-11-00739-t003:** The durations (minutes) of pre-surgery, surgery, and post-surgery.

Pre-Surgery	Surgery Case	Post-Surgery
Normal (8, 2)	Small: normal (33, 15)	Normal (28, 17)
	Medium: normal (86, 17)	
	Large: normal (153, 17)	
	E-large: normal (213, 17)	
	Special: normal (316, 62)	

**Table 4 healthcare-11-00739-t004:** The detailed information of four test cases.

Case No.	Number of Surgeries	PHU Beds	Operating Rooms	PACU Beds	Number of Surgery Types (S:M:L:E:SE)
1	10	2	3	2	2:6:1:1:0
2	15	3	4	3	3:9:2:1:0
3	20	3	4	4	4:12:3:1:0
4	30	4	5	5	7:18:3:1:1

**Table 5 healthcare-11-00739-t005:** Results of 10 surgeries.

Instance	10 Surgeries						
	LB	LPT			GA		
		$C_{max}$	CPU (s)	$(C_{max}-LB)/LB$	$C_{max}$	CPU (s)	$(C_{max}-LB)/LB$
1	359.33	387.00	0.04	7.70%	370.00	4.97	2.97%
2	349.33	378.00	0.04	8.21%	359.00	5.51	2.77%
3	354.67	386.00	0.03	8.83%	364.00	6.39	2.63%
4	374.00	389.00	0.03	4.01%	384.00	5.23	2.67%
5	362.33	400.00	0.03	10.40%	375.00	5.20	3.50%
6	324.67	351.00	0.03	8.11%	340.00	5.19	4.72%
7	355.33	381.00	0.04	7.22%	367.00	5.20	3.28%
8	374.33	422.00	0.03	12.73%	388.00	6.01	3.65%
9	345.00	386.00	0.03	11.88%	357.00	5.20	3.48%
10	346.67	409.00	0.08	17.98%	357.00	8.65	2.98%
Average	354.57	388.90	0.04	9.71%	366.10	5.75	3.27%

**Table 6 healthcare-11-00739-t006:** Results of 15 surgeries.

Instance	15 Surgeries						
	LB	LPT			GA		
		$C_{max}$	CPU (s)	$(C_{max}-LB)/LB$	$C_{max}$	CPU (s)	$(C_{max}-LB)/LB$
1	391.75	417.00	0.08	6.45%	407.00	11.60	3.89%
2	357.00	380.00	0.04	6.44%	373.00	7.57	4.48%
3	391.50	431.00	0.04	10.09%	408.00	9.94	4.21%
4	373.50	406.00	0.03	8.70%	392.00	7.09	4.95%
5	377.00	417.00	0.03	10.61%	396.00	13.39	5.04%
6	381.25	415.00	0.03	8.85%	397.00	11.84	4.13%
7	372.25	418.00	0.03	12.29%	388.00	6.72	4.23%
8	371.25	414.00	0.13	11.52%	389.00	7.13	4.78%
9	380.25	403.00	0.03	5.98%	397.00	6.86	4.40%
10	387.00	420.00	0.04	8.53%	407.00	6.65	5.17%
Average	378.28	412.10	0.05	8.95%	395.40	8.88	4.53%

**Table 7 healthcare-11-00739-t007:** Results of 20 surgeries.

Instance	20 Surgeries						
	LB	LPT			GA		
		$C_{max}$	CPU (s)	$(C_{max}-LB)/LB$	$C_{max}$	CPU (s)	$(C_{max}-LB)/LB$
1	505.25	541.00	0.03	7.08%	518.00	11.36	2.52%
2	492.75	516.00	0.04	4.72%	504.00	9.71	2.28%
3	494.25	533.00	0.04	7.84%	509.00	8.97	2.98%
4	484.00	518.00	0.04	7.02%	497.00	13.93	2.69%
5	494.00	541.00	0.03	9.51%	506.00	9.92	2.43%
6	499.25	521.00	0.04	4.36%	514.00	8.76	2.95%
7	473.25	493.00	0.09	4.17%	483.00	10.74	2.06%
8	520.25	543.00	0.03	4.37%	532.00	15.03	2.26%
9	482.25	503.00	0.04	4.30%	490.00	9.12	1.61%
10	472.00	515.00	0.09	9.11%	489.00	11.64	3.60%
Average	491.73	522.40	0.05	6.25%	504.20	10.92	2.54%

**Table 8 healthcare-11-00739-t008:** Results of 30 surgeries.

Instance	30 Surgeries						
	LB	LPT			GA		
		$C_{max}$	CPU (s)	$(C_{max}-LB)/LB$	$C_{max}$	CPU (s)	$(C_{max}-LB)/LB$
1	563.00	587.00	0.10	4.26%	577.00	16.40	2.49%
2	588.40	617.00	0.04	4.86%	604.00	17.61	2.65%
3	615.00	640.00	0.04	4.07%	626.00	27.51	1.79%
4	603.40	641.00	0.04	6.23%	622.00	13.49	3.08%
5	604.20	644.00	0.04	6.59%	619.00	17.82	2.45%
6	615.40	652.00	0.04	5.95%	629.00	19.66	2.21%
7	571.60	609.00	0.04	6.54%	586.00	16.74	2.52%
8	571.60	604.00	0.10	5.67%	585.00	13.62	2.34%
9	559.00	586.00	0.04	4.83%	573.00	14.80	2.50%
10	605.00	656.00	0.04	8.43%	619.00	15.36	2.31%
Average	589.66	623.60	0.05	5.74%	604.00	17.30	2.44%

**Table 9 healthcare-11-00739-t009:** The overall results of the GA.

Number of Surgeries	LB	LPT			GA		
		$C_{max}$	CPU (s)	$(C_{max}-LB)/LB$	$C_{max}$	CPU (s)	$(C_{max}-LB)/LB$
10	354.57	388.90	0.04	9.71%	366.10	5.75	3.27%
15	378.28	412.10	0.05	8.95%	395.40	8.88	4.53%
20	491.73	522.40	0.05	6.25%	504.20	10.92	2.54%
30	589.66	623.60	0.05	5.74%	604.00	17.30	2.44%
Average	453.56	486.75	0.05	7.66%	467.43	10.71	3.20%

## Data Availability

The data presented in this study is available on request from the corresponding author. The data are not publicly due to privacy.

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
