# Peer review of "Genetic Algorithm for Solving the No-Wait Three-Stage Surgery Scheduling Problem"

_healthcare, 2023, doi:10.3390/healthcare11050739_

Round 1

Reviewer 1 Report

This article tried to reduce the makespan of three-stage surgery using genetic algorithm. Although paper is easy to flow, however, I feel following major points must be addressed:

1. Introduction of this article needs a major revamp. You give to provide readers an idea about the shortcomings of makespan in three-stage surgery and its challenges. Based on these challenges, you can state contributions of your work and end it with organization of the paper.  Only contribution of the work can not be reducing makespan. List all contributions that allowed you to achieve this objective.  

2. I do not think why authors are explaining the feasibility of makespan in introduction. This is not the section where you will discuss feasibility of your approach.

3. I do not know what made authors to feel genetic algorithm might result in optimization. You need to state reasons for not using other algorithms and come with the details how GA suits your problem statement.

4. You need to compare evaluated results with state-of-the-art. You can not conclude your work by simply apply GA and LPT and show results in Tables. 

Author Response

Comments of Reviewer 1:

This article tried to reduce the makespan of three-stage surgery using genetic algorithm. Although paper is easy to flow, however, I feel following major points must be addressed:

  1. Introduction of this article needs a major revamp. You give to provide readers an idea about the shortcomings of makespan in three-stage surgery and its challenges. Based on these challenges, you can state contributions of your work and end it with organization of the paper.  Only contribution of the work can not be reducing makespan. List all contributions that allowed you to achieve this objective.  

We have stated the challenges of this work in Example A. We have also stated the contributions of our work and ended it with the organization of this paper in the introduction section.

  1. I do not think why authors are explaining the feasibility of makespan in introduction. This is not the section where you will discuss feasibility of your approach.

I have added a new section (section 3) where we provided a problem description. We also used a small example to explain feasible solutions for the example.

  1. I do not know what made authors to feel genetic algorithm might result in optimization. You need to state reasons for not using other algorithms and come with the details how GA suits your problem statement.

We have explained why we are not using other algorithms and why we thought GA is suitable for our problem in Section 2 literature review.

  1. You need to compare evaluated results with state-of-the-art. You can not conclude your work by simply apply GA and LPT and show results in Tables. 

To the best of our knowledge, there is no published research that studies the same problem as ours. We do not have state-of-the-art algorithms that we can compare directly. However, we have provided a detailed literature review on papers [18-21] that studied similar problems as ours but [18-21] also consider human resources and emergency surgeries. We also state the reason why we are not using mathematical model-based approaches as other researchers did and choose to use GA. Moreover, we have validated the effectiveness of the proposed GA by compared with the upper bound generated by the LPT and the lower bound [36]. The computational results indicate that the GA deviated from the lower bound by 3.25% on average and the average computation times of the GA is 10.71 seconds for surgery ranging from 10 to 30. We conclude that the GA can efficiently find near-optimal solutions for the daily three-stage operating room surgery scheduling problem.

Reviewer 2 Report

-Improve figure 1 resolution.

-The experiment setup is missing, so it is difficult to recreate the experiments the authors have done, specially in Section 3.

-Provide the bullet-wise "major contributions" for this paper at second last paragraph of the Introduction section. 

- Proposed algorithm is not self-explained. It is not clear which equations should be used to  enlarged sampling space (parents + offspring).It is recommended to revise the algorithm structuring into psudeo code.

- It is essential to make sure that the manuscript reads smoothly- this definitely helps the reader fully appreciate your research findings.

- It is recommended to draw one diagram/figure on recent advancement in proposed / existing in the Introduction Section.

-Add more references and cite below article to improve the readability of this paper:

(a) "Time-Series based Prediction for Energy Consumption of Smart Home Data Using Hybrid Convolution-Recurrent Neural Network", Naman Bhoj, Robin Singh Bhadoria, Telematics & Informatics (Elsevier), Vol. 75, p.101907, Dec 2022.

Author Response

Comments of Reviewer 2:

Comments and Suggestions for Authors

  1. -Improve figure 1 resolution.

We have improved figure 1 resolution.

  1. -The experiment setup is missing, so it is difficult to recreate the experiments the authors have done, specially in Section 3.

We have written the GA in pseudo code. So the reader should be able to implement the algorithm. The data are not publicly available due to privacy. However, the data presented in this study are available on request from the corresponding author.

  1. -Provide the bullet-wise "major contributions" for this paper at second last paragraph of the Introduction section. 

We have added the contribution of our work in the introduction section.

  1. Proposed algorithm is not self-explained. It is not clear which equations should be used to enlarged sampling space (parents + offspring).It is recommended to revise the algorithm structuring into psudeo code.

We have explained how to calculate the enlarged sampling space in section 4. We have also written the searching procedure of GA in pseudo code.

  1. It is essential to make sure that the manuscript reads smoothly- this definitely helps the reader fully appreciate your research findings.

We have reviewed the entire paper and checked grammar to ensure the manuscript reads smoothly.

  1. It is recommended to draw one diagram/figure on recent advancement in proposed / existing in the Introduction Section.

We are not quite sure what this comment means. We have added a typical patient flow in surgery and defined the scope of this research in the introduction section. We also give a literature review of recent articles that are related to this research.

  1. Add more references and cite below article to improve the readability of this paper:
    • "Time-Series based Prediction for Energy Consumption of Smart Home Data Using Hybrid Convolution-Recurrent Neural Network", Naman Bhoj, Robin Singh Bhadoria, Telematics & Informatics (Elsevier), Vol. 75, p.101907, Dec 2022.

We have cited the above article in the paper and we also add more references to improve the readability of this paper.

Reviewer 3 Report

The authors proposed GA based surgery to reduce time and resources for operating patients. Authors did a good job presenting and writing the manuscript. The reviewer has following concerns and questions about the article:

1. GAs are very old methods to optimize and there is plethora of scientific literature. Why did authors choose this method and not any ML-based optimization algorithm. Please describe it in the body of the paper as well as in review report.

2. The figures are not of high quality, please use only high quality .tiff figures.

3. To further improve the readability of this manuscript. I suggest adding contrasting literature from other domains in the introduction section and connecting to your proposed research on why GAs is better suited for your study and as whole research. 

Author Response

Comments of Reviewer 3:

Comments and Suggestions for Authors

The authors proposed GA based surgery to reduce time and resources for operating patients. Authors did a good job presenting and writing the manuscript. The reviewer has following concerns and questions about the article:

  1. GAs are very old methods to optimize and there is plethora of scientific literature. Why did authors choose this method and not any ML-based optimization algorithm. Please describe it in the body of the paper as well as in review report.

Only a few papers studied operating room scheduling problems in three stages [18-21]. In [18], the authors applied an ant colony (ACO) algorithm to solve a three-stage operating room scheduling problem integrating multiple nurses’ roster constraints. Similarly, in [19], the authors used an ACO to solve a three-stage operating room scheduling problem by considering available material resource (PHU beds, ORs, PACU beds) and human resources (surgeons, nurses, and anesthetists). Moreover, in [20], the authors proposed an iterative local search (ILS) and a hybrid GA (HGA) approach to solve the three-stage no-wait operating room surgery scheduling problem under various resource constraints. The availability of material resources, specialties, and qualifications of human resources are integrated, and the objective is to minimize makespan and total idle time in the operating rooms. They compared the proposed ILS and HGA with real data and the ACO algorithm [19]. The ILS and HGA outperformed the real data scheduling and the ACO scheduling [19] with close results and relatively short computation time. They allow fewer operating rooms and balance the working time between them. They concluded that the ILS and HGA algorithms are efficient and effective. In [21], the authors investigated operating room scheduling taking into account all resources, post-anesthesia beds, and emergency procedures. To solve the problem, they formulated a mixed linear integer programming (MILP) and transformed it into CP models. They also proposed a metaheuristic based on a GA and a constructive heuristic (CH). They compared the heuristic (GA combines with CH) with the MILP and CP models. They concluded that the CP model discovered high-quality solutions faster than the MILP. The. However, the improvement is not substantial. On the other hand, the. On the other hand, the heuristic found solutions of very good quality in short run time, with average of 7 seconds, on instances ranging from 15 to 40 surgeries requiring scheduling. The heuristic outperformed the MILP and CP models. Table 2 presents comparisons between key-related work and this study.

Some researchers tried to use mathematical model-based algorithms to solve operating room scheduling problems [6, 8, 15-17, 21]. In [6], the authors developed a branch-and-price approach to solve a surgeries assignment problem. In [8], the authors applied a column-generation-based heuristic to solve a tactical operating room planning problem. In [15], the authors proposed a primal–dual heuristic for solving an operation theatre planning problem. In [16], the authors presented a two-step approach to address an operating room scheduling problem with various resource-related constraints and the specifications of the operations processes. The two-step approach is based on solving mathematical models. In [17], the authors developed a constraint programming approach for solving multi-objective operating room scheduling. They considered multiple real-life constraints in the mathematical model, such as availability, staff preferences, and affinities among staff members. Mathematical model-based algorithms usually need advanced knowledge in math and optimization. It might not be easy to learn and implement for hospital managers. On the other hand, some researchers have recently addressed operating room scheduling problems using machine learning-based algorithms. Machine learning-based techniques are mainly employed to handle stochastic problems or problems with uncertain data. For example, [22] developed several machine learning techniques for predictive energy consumption data of smart residential homes. For operating room scheduling problems, machine learning-based algorithms are applied to predict surgery durations, to estimate required PACU time for each type of surgical procedure, to predict emergency patient arrival times, or detect surgeries with significant cancellation risks [23-27]. Since we are dealing with a deterministic three-stage operating room surgery scheduling problem, machine learning-based algorithms might not be suitable. Additionally, several researchers have applied metaheuristics to handle operating room scheduling issues, including GA [7, 12, 20-21, 28-29], ILS [20], ACO [18-19], and artificial bee colony (ABC) [10] algorithms. The advantage of metaheuristics is that they are easy to implement and can find good quality solutions within a reasonable time. That might better suit the needs of the hospital manager. In contrast to previous mathematical model-based approaches [6, 8, 15-17, 21], we aim to develop an algorithm that is easy to implement and flexible for the majority of hospitals. GA is one of the oldest and most well-known algorithms that are easy to learn and implement. Hence, we have chosen GA address the research problem.

  1. GA for a three-stage operating room scheduling problem

The GA was created by John Holland in 1960 [32]. GA is an adaptive technique that imitates the behavior of natural selection, evolution, and heredity. Although GA was created 60 years ago, it remains one of the most well-known and effective algorithms for solving various kinds of optimization problems. Readers interested in learning more about GA and its applications may refer to [33-34]. Many studies have successfully employed GAs to solve scheduling problems in operating rooms [7, 12, 20-21, 28-29]. The literature review demonstrates that GA can effectively address scheduling problems for operating room surgeries and deliver positive outcomes. GA is a good fit because we are dealing with a three-stage deterministic operating room surgery scheduling problem [20–21]. As a result, we created a GA to address the re-search challenge. 

  1. The figures are not of high quality, please use only high quality .tiff figures.

We have used high-quality .tiff figures throughout the whole paper.

  1. To further improve the readability of this manuscript. I suggest adding contrasting literature from other domains in the introduction section and connecting to your proposed research on why GAs is better suited for your study and as whole research. 

We have added contrasting literature from other domains in the literature review and connected it to our proposed research on why GA is better suited for our study and as whole research.

Round 2

Reviewer 1 Report

The revised version looks much improved. 

Reviewer 3 Report

The authors have improved the manuscript